# Investigation of Bioavailability and Food-Processing Properties of *Arthrospira platensis* by Enzymatic Treatment and Micro-Encapsulation by Spray Drying

**DOI:** 10.3390/foods11131922

**Published:** 2022-06-28

**Authors:** Patricia Maag, Simon Dirr, Özlem Özmutlu Karslioglu

**Affiliations:** 1Department of Food Technology and Horticulture, University of Applied Sciences Weihenstephan-Triesdorf, Am Hofgarten 4, 85354 Freising, Germany; patricia.maag@hswt.de (P.M.); simon.dirr@hswt.de (S.D.); 2Institute of Food Technology, University of Applied Sciences Weihenstephan-Triesdorf, Am Staudengarten 11, 85354 Freising, Germany

**Keywords:** *Arthrospira platensis*, microencapsulation, enzymatic treatment, digestibility, protein solubility, differential scanning calorimetry, colour intensity

## Abstract

Due to its high-protein content of 60–70% on dry weight, *Arthrospira platensis*, has been considered as one of the most sought-after protein alternatives. However, the processing of *Arthrospira platensis* extract (spirulina, SP) in food is usually limited due to the strong green colour and taste, as well as the lack of bioavailability of plant proteins. Therefore, this study aimed to increase its use in food applications through technologies such as microencapsulation by spray drying and enzymatic treatment. The effect of different combinations of maltodextrin (MD) and gum arabic (GA) as coating material were tested in ratios of 1:2 and 1:4 for *Arthrospira platensis*, core to wall material, respectively. Additionally, enzymatic treatment was used to investigate whether digestibility, protein solubility and powder solubility can be improved. Thermal stability was examined by differential scanning calorimetry (DSC), and colour intensity was analysed over L* a* b* colour system. The sample SP-MD1:2 showed the highest heat stability with a denaturation peak at 67 °C, while the samples SP-MD1:4 and ESP-MD1:4 revealed the best brightening effects. The crude protein content stated by the manufacturer of 67% was confirmed. Encapsulation and enzymatic hydrolysis enhance the protein solubility, under which ESP-MD1:4 had the greatest solubility at around 83%. The protein digestibility peaks were around 99% with sample SP-MD1:2.

## 1. Introduction

The world’s population is continuously growing and reached roughly 7.9 billion people in 2021 [1]. At the current rate of growth, we will reach a world population of 9.7 billion in 2050, according to the United Nations [2]. With respect to this growing world population, the considerable amount of 8.9% (690 million people) is hungry [3]. In contrast, in well-industrialized countries, people are facing various needs regarding food consumption. Socio-economic changes such as ethical concerns, healthy diets like protein-rich nutrition, higher income, and urbanization have increased the demand for healthy and high-quality functional foods based on natural and sustainable origins and cultivation [4,5]. This out-of-balance supply and demand for “high-quality” food has driven us to consider how to serve the growing world with protein-rich food and led us to develop more sustainable plant-based food products with a balanced nutrient composition, especially for meat and dairy alternatives. In addition, environmental influences also play a major role. Notably, some other studies have calculated that a 50% substitution of meat with plant-based proteins can reduce greenhouse gas emissions by 32%. More precisely, the carbon footprint for plant-based food is on average twice as low as, for example, pork production [6]. For the above-mentioned reasons, the use of alternative proteins such as algae (spirulina and chlorella), insect protein, or cultured meat has become the focus of researchers and is of great importance to providing a sustainable and protein-rich diet [7]. As an innovative protein alternative, *Arthrospira platensis* (spirulina, SP), with up to a 65% protein content in dry matter, has become of increasing interest in food enrichment in the industry of European countries [4].

Spirulina is a multicellular, helical and filamentous cyanobacterium that shows a characteristic blue-green colour. Microalgae are estimated to have already been on Earth about 3.6 billion years ago [8], and they are probably the ancestors of plants [9]. They are exceptionally rich in valuable nutrients such as minerals, especially calcium (333 mg/100 g), magnesium (500 mg/100 g), and iron (217 mg/100 g) [10]. Additionally, they are rich in vitamins (Vit. B1, B2, B12, E) and omega-3-fatty acids such as linolenic acid and are found to be ideally balanced as complete food. With respect to their high protein content, it provides the full range of essential amino acids, which is why it has become of special interest as plant-based protein source [8].

Spirulina, as a potential protein alternative, is not highly demanding and is cost-effective in terms of cultivation. The optimum growth conditions are at temperatures ranging from 29–35 °C and a pH between 9 and 11, with 30% of the day being spent in full sunlight to ensure healthy growth. For spirulina cultivated indoors, regular agitation and aeration are essential, requiring homogeneous distribution of carbon dioxide-enriched airflow, with a required CO_2_ concentration of 0.5%, ultimately leading to optimal productivity and good quality in terms of biomass density and yield [8].

Nevertheless, spirulina, as a new food source brings some challenging characteristics for processing in food products. The strong, dark, green-blue colour, coming from pigments chlorophyll a, phycocyanin, and carotenes, is a challenging factor which limits its usage in food products. Spirulina has proven antioxidant and anti-inflammatory effects on the human body and is therefore available on the market as a food supplement in the form of powder and tablets [10], but it has so far only been added to foods to a limited extent for protein enrichment, e.g., in protein bars or smoothies. The limiting factors are the loss of taste and the strong green-blue colour, both of which restrict product development. Therefore, technologies such as microencapsulation by spray drying and enzymatic treatment have become of interest within this study to overcome spirulina’s restricted use in various food products.

The aim of this study was to formulate a spirulina extract with reduced colour intensity, optimized thermal stability, increased solubility, and moreover improved digestibility through the application of enzymatic treatment and microencapsulation. For this purpose, microencapsulation by spray drying was chosen as a method to entrap sensitive nutrients in order to protect them from physico-chemical influences and external factors such as heat and oxidative processes caused by air and light. Suitable wall materials maltodextrin and gum arabic were selected. Da Silva et al. (2019) has used microencapsulation by spray drying on spirulina extract to improve food formulations. Here, the enrichment of spirulina extract in yogurt was enabled and found to have both a positive effect on the emulsification properties and to result in a more homogeneous and less green appearance than pure, non-encapsulated spirulina powder [11]. Moreover, spirulina extract encapsulated by spray drying technology with a polysaccharide such as maltodextrin was proven to have increased thermal stability [11,12]. In addition, polysaccharides are also excellent for masking unpleasant and characteristic odours, which was investigated by Lan et al. (2019) for isolated pea proteins [13]. Furthermore, enzymatic treatment with proteases can optimize the bioavailability and thus the digestibility of plant proteins and offer the possibility of colour reduction, which is being investigated in connection with microencapsulation.

## 2. Materials and Methods

### 2.1. Materials

Spirulina extract (*Arthrospira platensis*) with about 66% protein content was obtained from the enterprise Bionutra GmbH, Germany, and was delivered in a grinded powder formulation for further processing. For protein hydrolysis, Biocatalysts Ltd., Cardiff, UK, kindly provided the enzyme Promod 184 MDP. The enzymes pepsin from Carl Roth GmbH & Co., KG (Karlsruhe, Germany), and pancreatin from Merck KGaA, Darmstadt, Germany, were used to measure protein digestibility. The wall materials for microencapsulation, such as maltodextrin, were purchased from Biozoon GmbH, Bremerhaven, Germany, and gum arabic was from Naturix24 GmbH, Dransfeld, Germany. For the performance of the analysis, all other used chemicals and reagents were of analytical grade and commonly used laboratory equipment was utilized.

### 2.2. Sample Preparation

#### 2.2.1. Enzymatic Treatment

A suspension with a concentration of 4% (*w/w*) spirulina extract and distilled water was prepared and mixed at 40 °C for 20 min in a multifunctional processor using the thermomixer TM5 (Vorwerk, Germany) to ensure a solvent-to-dry-matter ratio of 1/20 and sufficient hydration based on previous trials. The prepared suspension was then homogenized at 200 bar by the two-stage homogenizer Panda Plus 2000 (GEA Niro Soavi, Italy) to induce cell disruption, which leads to greater accessibility of proteins and improved enzymatic hydrolysis of released proteins. Enzymatic hydrolysis was carried out by adding 2% (*w/w*) of Promod 184 MDP (Biokatalyst Ltd., Cardiff, UK) based on the protein weight, and the spirulina extract was enzymatically treated for 60 min at an optimum temperature of 55 °C and a pH of 7. The subsequent spray drying process inactivated the enzymatic treatment.

#### 2.2.2. Microencapsulation by Spray Drying

Microencapsulation by spray drying was conducted using a Mini-Spray-Dryer B-290 (Büchi GmbH, Flawil, Switzerland). The B-290 works on the principle of direct current atomization by a two-substance nozzle. Microencapsulated samples were prepared as mentioned in 2.2.1. To be more precise, enzymatically treated spirulina extract (ESP) and non-enzymatically treated spirulina extract (SP) were mixed with either maltodextrin or a mixture of maltodextrin/gum arabic (1:1) in respective ratios of 1:2 and 1:4 (*w/w*), core to wall material. As reference, a non-encapsulated spirulina extract (SP4) and a non-encapsulated enzymatically treated spirulina extract (ESP4), each prepared as a 4% (*w/w*) suspension, were subjected to the same spray-drying procedure for comparison. Before spray drying, each suspension was homogenized at 200 bar by a two-stage homogenizer (Panda Plus 2000 [GEA Niro Soavi, Parma, Italy]) to guarantee a homogenous suspension with fine dispersed particles.

The spray drying process was also carried out using the spray dryer B-290 (Büchi, Flawil, Switzerland). The spray drying process was selected to perform optimal drying results for low water activity, high product yield, and low thermal load. Therefore, the inlet temperature was set to 120 °C, and the resulting outlet temperature reached 70 °C at a spray flow of 0.667 m^3^/h. The aspirator was set at 90% to achieve the best separation of dried particles and hot humid air, and the pumping capacity was about 18–20%. The obtained samples were collected in containers and stored at 4 °C until further analysis.

### 2.3. Powder Solubility

The solubility of enzymatically treated and spray-dried samples as well as untreated samples was determined according to the patented methodology from Eastman et al. [14] with some modifications. Samples, of 0.5 g of each powder, were dissolved in 50 mL of distilled water and stirred for 10 min at room temperature at 200 rpm. The samples were then centrifuged at 800 rpm for 5 min. The supernatant, with a liquid quantity of 25 mL for each sample, was taken and transferred into an aluminium tray and then kept at 110 °C for 18 h in a drying oven until complete evaporation of the water. Finally, the dried samples were put in a desiccator for cooling down to ambient temperature. The powder solubility was expressed in percentage (%) according to the following equation:(1)$$\%\;powder\;solubility=\frac{weight\;of\;dried\;solids\;\left[g\right]\times 2}{total\;weight\;of\;sample\;\left[g\right]}\times 100$$

### 2.4. Water Activity

Water activity was measured using the LabMaster aw (Novasina GmbH, Kempen, Germany). An appropriate amount of 1 g of each sample was evenly spread on a plastic tray, and the result was given after reaching the moisture equilibrium. The water activity was used to investigate the storage stability of the spray-dried powder.

### 2.5. Thermal Stability and Protein Denaturation

The heat stability of the samples was determined by differential scanning calorimetry (DSC) using a DSC Q200 (TA Instruments, New Castle, DE, USA) equipped with a cooling system using nitrogen gas. With the DSC measurement, the denaturation temperature (T_d_), which is referred to as ‘thermal reactivity’ to infer heat stability, of proteins from untreated samples and treated (enzymatic, microencapsulated) samples was investigated. The calibration was carried out with indium. High volume pans made of stainless steel, with a filling volume of 100 μL and which can withstand a maximum pressure of 35 bar, were used for DSC measurement. A solid solution of 20% from each powder sample was prepared with distilled water under continuous stirring for 15 min, 30 mg samples of which were taken, containing 4.4% or 2.64% protein at ratios of 1:2 and 1:4 (*w/w*), respectively. The pH value of each sample was kept at neutral and ranged from 6.5–6.9. An empty pan filled with distilled water was used as reference. The experiment was performed within a temperature range of 30–150 °C at an underlying heating rate of 2 K/min for each sample. Each measurement was conducted in triplicate.

### 2.6. Colour Intensity

Digital capturing and visualization of colour differences was performed using a DigiEye (VeriVide, Leicester, UK) in diffuse mode to remove specular reflection. A standardized light source CIE D65 was chosen, as most food is viewed in daylight. The angle of observation was set to 10° to simulate a wide field of view. The L* a* b* values were determined, which stand for coordinates in the CIELAB colour room. When comparing two different colours, differences in the colour coordinates are formed. These are marked with delta ∆. Determination of the total distance, ∆E, can be calculated via Equation (2). The ∆E indicates the total colour difference without direction.
(2)$$\Delta E=\sqrt{{\left(\Delta L^{*}\right)}^{2}+{\left(\Delta a^{*}\right)}^{2}+{\left(\Delta b^{*}\right)}^{2}}$$

### 2.7. Particle Size Distribution

A particle size analyzer S3500 from Microtrac (Montgomeryville and York, PA, USA) was applied using the laser diffraction technique, consisting of a “tri-laser-system” to determine the particle size distribution. The S3500 was equipped with a dry-dispersion module operating in a measuring range between 0.1–1000 µm. The results of the particle size distribution were displayed as a cumulative distribution function. To characterize the cumulative distribution, the values d10, d50, and d90 were determined. For this purpose, the percentage volume fraction of the particles was plotted or summed over the particle size dx-. The value dx represents the particle size diameter, where 10, 50 or 90% of the volume of particles has a smaller diameter.

### 2.8. Protein Solubility

The protein solubility of the enzymatically treated spirulina extract was determined by the method of Morr et al. (1985) with slight modification [15]. A quantity of 1 g with a 0.001 g accuracy was weighed into a 100 mL beaker and mixed with 40 mL of 0.1 mol/L sodium chloride solution for 30 min under continuously stirring. The suspension was adjusted with either 0.1 mol/L sodium chloride solution or 0.1 mol/L sodium hydroxide solution to a desired neutral pH value (pH 7) and was then transferred into a 50 mL volumetric flask and filled up with 0.1 mol/L sodium chloride to the defined quantity. An appropriate amount of 20 mL was transferred into a centrifuge tube and was centrifuged for 20 min at 4000 rpm. The resulting supernatant was filtered with a Whatman No. 1 filter, and 10 mL of the supernatant was taken to measure the protein content via Kjeldahl; it was calculated to have a protein factor of 6.25. Finally, as shown in Equation (3), the protein solubility was determined by relating the protein content of the supernatant to the protein content of the initial sample.
(3)$$\%\;protein\;solubility=\frac{protein\;content\;of\;supernatant\;\left[g\right]}{protein\;content\;of\;initial\;sample\;\left[g\right]}\times 100$$

### 2.9. Digestibility

The analysis of digestibility was conducted according to the method of Kaur et al. (2010) [16] with slight modification. Each sample, with an amount equivalent to 70 mg nitrogen, was transferred into a 50 mL centrifuge tube containing 17 mL of 0.1 M hydrochloric acid and was mixed at 37 °C for 30 min under continuous stirring. In the next step, the solution was adjusted to a pH value of 1.9, and 2.5 mL pepsin solution (2.5 mg/mL dissolved in 0.1 M HCL solution) was added to the mixture and incubated for one hour. After the first incubation period, the pepsin was inactivated by increasing the pH value up to 8, and a second incubation period of two hours with 2.5 mL pancreatic solution (2.5 mg/mL dissolved in 0.1 M sodium phosphate buffer) was conducted. After two hours, the reaction was stopped with 5 mL trichloroacetic 10%, and the suspension was centrifuged at 4000 rpm for 5 min. With a volumetric pipette, 10 mL of the supernatant was examined for protein content with the method of Kjeldahl. The protein digestibility was calculated by relating the determined protein content of the supernatant to the protein content of the initial sample, which can be seen in the following equation:(4)$$\%\;Protein\;Digestibility=\frac{\frac{P_{supernatant\;\left[\mathrm{mg}\right]}}{10\;\mathrm{mL}}}{\frac{m_{sample\;\left[\mathrm{mg}\right]}\times P_{sample\;\left[\%\right]}}{V_{total\;sample\;\left[\mathrm{mL}\right]}}}$$

### 2.10. Statistical Analysis

All experiments were performed in triplicate unless otherwise noted. Mean values are presented with standard deviations and are compared using a one-way ANOVA with an enclosed Tukey’s honestly significant difference (HSD) test using testing significance α = 0.05. 

## 3. Results and Discussion

### 3.1. Physical Parameters of Enzymatically Treated and Spray-Dried Arthrospira platensis

#### 3.1.1. Water Activity (a_w_)

Water activity is a measure of the availability of free water, i.e., unbound water that is not linked with ions (e.g., Na^+^/Cl^−^) or sugar molecules. In the food industry, the measurement of the water activity is used for quality control to ensure the durability of products. Within the scope of this investigation, information is to be obtained on whether the spray drying process runs with a constant drying performance, so that the spray drying process can be adapted in terms of drying performance for processing and resulting storage stability.

The water activity values of the two references, SP4 and ESP4, did not differ significantly. Samples spray dried in the ratio of 1:2 (*w/w*), core to wall material, show no significant differences, while the significant differences between some samples spray dried in ratio 1:4 (*w/w*), core to wall material, are presented in Table 1. Overall, the water activity from all spray-dried samples was in the range of 0.15 to 0.21. Similar water activity values (0.28), and even slightly higher values for spray-dried soybean hydrolysate in maltodextrin and soy protein isolate, were shown by Wang et al. (2020) [17].

Enzymatic treatments did not influence the drying process significantly. The differences in water activity values are more likely due to slight fluctuations in the drying process and could be due to instability in the parameter settings of the Büchi B-290 spray dryer. Spray drying on a lab scale demands continuous control and adjustment of the settings, especially for differently composed samples, in contrast to an industrial spray dryer. However, the water activity values from all samples were within the expected range of powdered products. All samples were clearly below the recommended value of 0.6 and would not be susceptible to microbial infestation or to yeasts and molds [18].

#### 3.1.2. Thermal Stability and Protein Denaturation

For preservation and food processing, heat treatment is one of the most important processing steps to guarantee a certain shelf life, and in addition to other characteristics, textural properties can be influenced. Therefore, heat-induced conformational changes to the present microencapsulated proteins from spirulina powder were analysed using the DSC (differential scanning calorimetry) method, whereby the protein denaturation as well as the thermal stability of microcapsules was observed. Furthermore, the impact of enzymatic treatment and microencapsulation using the wall materials maltodextrin and gum arabic on heat stability and protein denaturation was investigated. The polysaccharide maltodextrin and the hydrocolloid gum arabic were chosen as wall materials because of their ability to interact with proteins. Cross-linking is generated between the carboxyl groups from proteins and hydroxyl groups from polysaccharides, thus enabling stable compounds [19]. In a modified status, maltodextrin (DE19), as used in this study, shows thermal decompositions at around 290 °C [20] with water activities of 0.75, and for gum, thermal decompositions are presented at 270 °C [21]. In addition, gum arabic as encapsulation material reveals even better film-forming properties but lacks in solubility. Therefore, in recent studies, gum arabic was mixed with maltodextrin, which delivers excellent solubility properties and additionally provides good heat stability [20].

The denaturation peaks of spirulina reference (SP4) seen in Table 1 exhibited two endothermic peaks with midpoint transition temperatures at 64.7 and 108.9 °C. This is very much in accordance with Chronakis et al. (2001), who investigated spirulina protein isolate, observing two denaturation peaks at 67.0 and 109.0 °C [22]. Denaturation temperature, focusing on the first peak, from non-enzymatically treated but microencapsulated samples ranged from 65.4 to 67.3 °C. It is noticeable that the samples SP-MD1:4 and SP-MDGA1:4 did not show a second peak but showed a first peak at 54.4 and 67.3 °C, respectively. The samples SP-MD1:2 and SP-MDGA1:2 instead revealed two peaks, the first peaks at 67.2 and 66.2 °C and the second peaks at 109.7 and 108.5 °C, respectively. It is possible to assume that the amount of protein at a ratio of 1:4 (*w/w*), which contains roughly about 0.88% spirulina powder, indicates a value which is below the detection level. Another reason could be that proteins are already denatured or aggregated due to previous processing steps or spray drying supporting the statement of Tanger et al. (2020) and Taherian et al. (2011) [23,24]. Further direct correlations are observed between water activity values and denaturation peaks. An increased water activity led to a decrease in denaturation temperature. The sample SP-MD1:4 showed the lowest T_d_ at 65.4 °C and the highest water activity of 0.19 for all samples. It is notable that all other non-enzymatically treated but microencapsulated samples, including the reference SP4, SP-MDGA1:2, SP-MD1:2, and SP-MDGA1:4, respectively, represent descending a_w_ values and ascending denaturation temperatures. Daza et al. (2016) made similar observations for spray-dried powder extracts [25]. Overall, the non-enzymatically treated but microencapsulated samples showed significantly higher denaturation temperature peaks, except for SP-MD1:4, but not to the extent expected.

Enzymatically treated spirulina powder, referred to as reference ESP4, did not give any thermal peaks, most likely due to peptide cleavage by the enzymes used. Peptides are too small to be detected and thereby demonstrate no thermal impacts.

No significant peaks could be detected in any encapsulated samples that had previously been subjected to enzymatic treatment. This is in line with the already mentioned observation that was made for the reference sample ESP4. There is reason to believe that the peptide bonds of proteins have been cleaved by enzymes to such an extent that they are insignificant and deliver no visible peaks. Costa da Silva et al. (2021) observed similar results for hydrolysates from microalgae spirulina sp. LEB 18 and also used proteolytic enzymes. Due to enzymatic hydrolysis, the conformational structure of proteins was changed by destruction of the quaternary and tertiary structure. This might lead to peptide fractions with lower molecular weight and more stable amino acids which are more heat stable. Costa da Silva et al. (2021), who detected the first temperature-induced decompositions at 180 °C for hydrolyzed spirulina, confirmed this assumption. For this reason, mass loss as a function of temperature induced higher temperature peaks, which may have exceeded the maximum set temperature within DSC measurement of 150 °C in this study [26].

#### 3.1.3. Colour Intensity

The colour of food is one of the distinguishing attributes that consumers use as an indicator for quality. A non-appealing colour can determine the liking or disliking of one product over another. Therefore, colour measurement is implemented as a quality control method in the food industry. In this study, the focus is to receive a nearly white, fine powder through microencapsulation and enzymatic treatment combined with heat treatment. The optimum values for a white powder are L*: 100, a*: 0, and b*: 0. A neutral colour helps tremendously for applications in beverages. The green-blue colour of spirulina might be a hurdle for its use in drinks. The L*, a*, b* values of the spray-dried spirulina powder are displayed in Table 1 and are visually illustrated in Table 2. The reference was represented through sample SP4, and the mean is compared to every treated sample. The total colour difference is displayed via the ∆E value. In general, microencapsulation as well as enzymatic treatment had a significant impact on colour. Microencapsulation had the greatest impact in brightening the powder and increasing the L* value by approximately 10–13% for the ratio 1:2 (SP-MD1:2) and 20% for the ratio 1:4 (SP-MD1:2). An enzymatic treatment had an impact of nearly 2% (ESP) on the L* value. Enzymatic treatment had the most impact on the a* and b* value. The a* value was increased by 10 to 15% for the samples ESP-MD1:4, ESP-MD1:2, ESP-MDGA1:4, ESP-MDGA1:2, and ESP. ESP had the best results at the a* level, which led to a brighter green colour. This can be linked to chlorophyll destruction due to thermal processing. The b* value increased 3–6% with enzymatic hydrolysis, which led to less blue and s greater yellow impression. Encapsulation, on the other hand, decreased the b* value by around 2–5%, which lead to a less yellow and more blue impression compared to SP4.

#### 3.1.4. Particle Size and Powder Solubility

The particle size distribution and powder solubility of the microencapsulated samples were measured to compare correlations between particle size and solubility and the influence of enzymatic pre-treatment. In addition, the particle size distribution provides information on whether the spray-dried particles are largely uniform or whether agglomeration has occurred due to process variations. Larger particles can therefore influence the solubility of the powder, which in turn affects further food processing steps.

The total particle size distribution of all samples, which can be seen in Table 3, is between 1.8–29 µm and is a typical range for spray-dried powders. Particle sizes should not exceed 50 µm, since they can significantly influence the texture and sensorial feel [27]. For example, Tan et al. (2020) microencapsulated whey protein isolate by spray drying with resulting particle sizes that range from 3–30 µm. The results confirmed that smaller particles increased in size by 50% over a given gel time, which is related to the formation of a larger surface area by smaller particles, and consequently, more water can be absorbed [28].

In the current study, particle size for non-enzymatic but microencapsulated samples showed smaller particles than for the additionally enzymatically treated samples. The same pattern was also shown by the two samples SP4 and ESP4, with SP4 revealing that, for 90% (d90) of its particle volume, the particles were smaller than those of ESP4. The two different ratios used, 1:2 and 1:4 (*w/w*), did not show differences in particle sizes among themselves. However, it can be said that the combination of two wall materials, maltodextrin and gum arabic, compared to use of only maltodextrin. slightly increased the particle size for non-enzymatically treated microencapsulated samples, but this does not apply to those that were enzymatically treated.

The occurrence of larger particles in the enzymatically treated samples might be explained by agglomeration of the proteins via protein–protein bonds. Hydrolysis by enzymes can cause the unfolding of proteins, and subsequently, aggregations result in larger particles. Tamm et al. (2016), who microencapsulated pea protein hydrolysate via spray drying, confirmed this observation [19].

A direct correlation could not be observed between particle size and solubility. As shown in Figure 1, all microencapsulated samples showed significantly improved solubility compared to the references SP4 and ESP4. The non-enzymatically treated microencapsulated samples showed, on average, a 44% increased solubility, while the solubility of enzymatically treated encapsulated samples improved by 25%. It should be taken into account that the references SP4 and ESP4 resulted in a respective powder solubility of 46% and 65%. There were only little variations among results of all the microencapsulated samples and all samples showed overall increased solubility. However, no statement can be made that the ratios lead to better solubility. The additional use of gum arabic indicates a slight deterioration in solubility. The best solubility results were achieved with SP-M1:2 and ESP-MD 1:4, with solubilities of 92.68 and 93.65%, respectively. Similar results for solubility were investigated by Wang et al. (2020), who spray dried soy protein hydrolysate with maltodextrin and reached solubilities between 94–97% [17]. Maltodextrin and gum arabic are therefore very well suited as carriers for the microencapsulation of proteins, as they complement each other very well. Maltodextrin has very good solubility properties, which are less present in gum arabic, which in turn contributes with its great film-forming ability, an important factor for encapsulation [19]. It was also found that interactions between proteins and carbohydrates lead to stable emulsions and provide better protection against the oxidation of active nutrients, making microencapsulation interesting for the food industry [29].

### 3.2. Bioavailability of Spray-Dried Microcapsules

#### 3.2.1. Protein Content

The protein content was measured to validate the manufacturer’s claims and was used for further calculations such as protein solubility and digestibility. The manufacturer’s claim that the raw material contained 67% protein was confirmed. Table 4 displays that the enzymatically hydrolyzed samples (ESP4) had a lower protein content. This can be explained due to the losses at the drying chamber during the spray drying process. The encapsulated samples had a lower protein content due to the reduced amount of raw material. Considering the ratio of core to wall material, the values correspond to the raw material’s protein content.

#### 3.2.2. Protein Solubility

According to the literature, *Arthrospira platensis* protein isolate has its lowest protein solubility at pH 3 (6.2%) and its maximum at pH 10 (59.6%), which validates the generated data for SP4 (53.9% at pH 7) [30]. The results in Table 4 indicate that ESP4 had a significantly higher protein solubility of approximately 10% compared to SP4 due to a greater amount of smaller peptides [31]. Therefore, a treatment with proteolytic enzymes was sufficient to increase the protein solubility.

Any performed encapsulation also improved the protein solubility significantly compared to SP4 and ESP4, with an exception for ESP4 and SP-MDGA1:4. This suggests that encapsulation with or without enzymatic hydrolysis increased the solubility of the protein, but this can be retained with higher amounts of gum arabic [20].

ESP-MD1:4 had the highest protein solubility: 83%. SP-MD1:2 and SP-MD1:4 had a solubility value of around 79%, which was 4% lower than ESP-MD1:4. Considering the factors determining the feasibility of the process (time, price for enzymes, more processing), SP-MD1:2 and SP-MD1:4 provided the most effective solution to increase the protein solubility. Further research must be done to validate those statements for more insights on the influence toward functional properties.

#### 3.2.3. Protein Digestibility

Measuring the digestibility of a product can give some indication of its bioavailability. Therefore, in vitro methods were used to estimate the bioavailability of the samples. Results for the protein digestibility are displayed in Table 4. The digestibility of SP4 had no significant difference compared to ESP. The obtained digestibility values were 58.68% for SP4 and 60.47% for ESP. This is contrary to expectations that the enzymatically treated sample, which had a greater protein solubility, had a similar digestibility. According to the literature, freeze-dried powder of untreated *Arthrospira platensis* had a digestibility of around 65% [32]. The difference with the results of this study can be explained by the different drying methods. Hsu et al. (1977) points out the possibility of decreased digestibility induced by Maillard reaction through thermal processing or thermal cross-linking of proteins [33].

Similar to the results for the protein solubility, any encapsulation with or without enzymatic treatment enhanced the protein digestibility compared to SP4 and ESP4. ESP-MD1:2 had the best digestibility at around 90%. SP-MD1:2, SP-MD1:4, and SP-MDGA1:2 had no significant difference with a value around 80%, in addition to SP-MDGA1:4 and ESP-MDGA1:2 with a value around 83%. Within the enzymatically treated and encapsulated samples, there was a tendency that suggests the encapsulation ratio of 1:2 is favorable.

## 4. Conclusions

Microencapsulation and additional enzymatic hydrolysis were applied to spirulina extract in order to reduce the intense green colour, to increase bioavailability, and to enhance protein and powder solubility.

Optimized spray drying conditions delivered a beneficial particle size distribution of the microencapsulated powder (18–29 µm) for the final food applications. Microencapsulation brought a significantly higher denaturation temperature for the majority of treated samples. As one of the major outcomes, a considerable reduction in green colour intensity was observed for encapsulated samples. That reduction effect will be more dominant if the protein source is isolated. Results showed as well that enzymatic treatment or encapsulation enhances the protein solubility between 10 and 30% compared with the untreated samples. Further research could distinguish the influence of encapsulation and/or enzymatic treatment for further functional properties and specific applications. Furthermore, powder solubility and digestibility were also enhanced significantly. A 20% increase in protein digestibility can be reached through encapsulation and enzymatic hydrolysis. Further research could focus on more purified concentrates or isolates to see how this enhancement can be implemented for food processing benefits.

## Figures and Tables

**Figure 1 foods-11-01922-f001:**
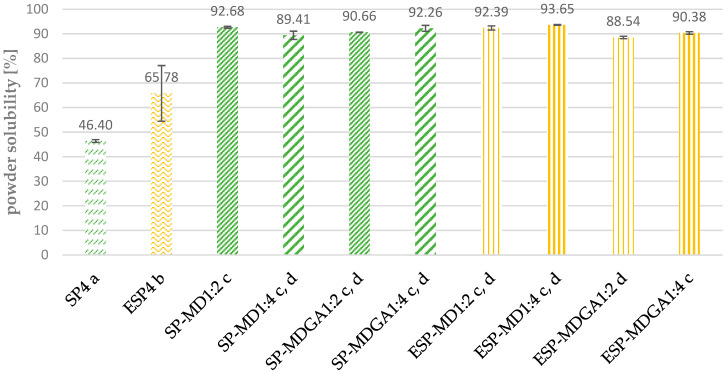
Powder solubility of microencapsulated compared to non-encapsulated *Arthrospira platensis*. Values with the same lowercase letters (a–d) did not show any significant difference at *p* < 0.05, according to Tukey’s HSD test.

**Table 1 foods-11-01922-t001:** Physical properties of microencapsulated *Arthrospira platensis*.

Sample	Water Activity *	Thermal Stability [°C] *		Colour Intensity *			
		T_d1_	T_d2_	L *	a *	b *	∆E
SP4	0.187 ± 0.003 ^a,b^	64.7 ± 1 ^a^	108.8 ± 0.18 ^a^	35.34 ± 0.99 ^j^	−21.57 ± 0.64 ^a^	19.40 ± 0.41 ^a^	0
ESP4	0.212 ± 0.004 ^a^	n.d.	n.d.	37.89 ± 1.10 ^i^	−5.84 ± 0.29 ^e^	25.47 ± 0.58 ^b,c^	+17.04
SP-MD1:2	0.163 ± 0.00 ^b^	67.2 ± 0.10 ^b^	109.7 ± 0.35 ^b^	44.46 ±0.14 ^h^	−19.05 ± 0.43 ^b,c^	17.80 ± 0.26 ^d,e^	+9.58
SP-MD1:4	0.194 ± 0.017 ^a,c^	65.4 ± 0.16 ^a,c^	n.d.	55.62 ± 0.08 ^a,b,c^	−17.92 ±0.42 ^b^	17.83 ± 0.24 ^d,f^	+20.46
SP-MDGA1:2	0.179 ± 0.003 ^b,c,d^	66.2 ± 0.10 ^b,c^	108.5 ± 0.17 ^a^	48.35 ± 0.26 ^d^	−21.28 ± 0.47 ^a^	18.16 ± 0.75 ^a,e,f^	+12.87
SP-MDGA1:4	0.154 ± 0.001 ^b^	67.3 ± 0.51 ^b^	n.d.	54.41 ± 0.00 ^a,e,f^	−19.57 ± 0.00 ^c^	15.95 ± 0.00 ^i^	+19.47
ESP-MD1:2	0.168 ± 0.014 ^b,e^	n.d.	n.d.	50.26 ± 0.00	−9.34 ±0.00 ^d^	25.47 ± 0.00 ^b,g^	+20.04
ESP-MD1:4	0.155 ± 0.013 ^b^	n.d.	n.d.	55.77 ± 0.15 ^b,e,g^	−11.69 ±0.39 ^f^	23.50 ± 0.23 ^h^	+23.05
ESP-MDGA1:2	0.152 ± 0.006 ^b^	n.d.	n.d.	48.61 ± 0.14 ^d^	−7.24 ± 0.34 ^g^	25.25 ± 0.19 ^c,g^	+20.37
ESP-MDGA1:4	0.194 ± 0.017 ^a,d,e^	n.d.	n.d.	54.78 ± 0.01 ^c,f,g^	−9.19 ± 0.05 ^d^	22.13 ± 0.39 ^h^	+23.23

* Values with the same lowercase (^a–j^) letters in the same columns did not show any significant difference at *p* < 0.05, according to Tukey’s HSD test. n.d. = not detectable.

**Table 2 foods-11-01922-t002:** Colour intensities of powder samples.

SP4 *	ESP4 *	SP-MD1:2 *	SP-MD1:4 *	SP-MDGA1:2*
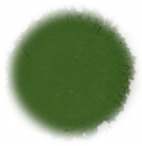	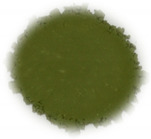	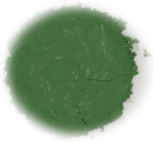	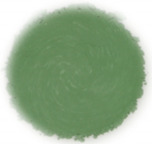	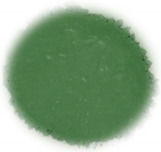
SP-MDGA1:4 *	ESP-MD1:2 *	ESP-MD1:4 *	ESP-MDGA1:2 *	ESP-MDGA1:4 *
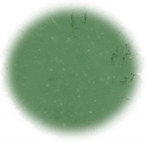	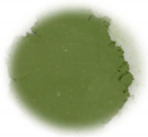	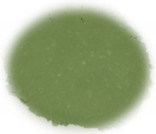	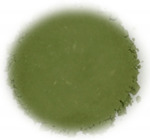	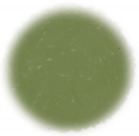

* SP4: spirulina powder 4% (*w*/*w*), ESP4: enzymatically treated spirulina 4% (*w*/*w*) used as core materials; MD: maltodextrin, GA: gum arabic used as wall materials, either separately or in combination MDGA (50:50); Selected ratios core to wall material 1:2 or 1:4.

**Table 3 foods-11-01922-t003:** Particle size distribution of microencapsulated *Arthrospira platensis*.

Sample	Particle Size in %Tiles *		
	d10 [µm]	d50 [µm]	d90 [µm]
SP4	1.8 ± 0.13 ^a^	6.7 ± 0.2 ^a^	13.36 ± 0.72 ^a^
ESP4	2.22 ± 0.02 ^b^	6.98 ± 0.02 ^a,c^	17.55 ± 1.06 ^b^
SP-MD1:2	1.75 ± 0.01 ^a^	7.09 ± 0.06 ^b,c^	14.75 ± 0.36 ^a^
SP-MD1:4	1.85 ± 0.01 ^a,c^	7.37 ± 0.11 ^b^	15.55 ± 0.8 ^a,b^
SP-MDGA1:2	3.07 ± 0.01 ^d^	11.17 ± 0.07 ^d,g^	24.44 ± 0.65 ^c,d^
SP-MDGA1:4	2.06 ± 0.02 ^b,c^	7.85 ± 0.07 ^e^	17.98 ± 0.49 ^b^
ESP-MD1:2	3.01 ± 0.02 ^d^	10.84 ± 0.01 ^d,f^	22.76 ± 0.18 ^c,d^
ESP-MD1:4	3.03 ± 0.09 ^d^	11.37 ± 0.06 ^d,g^	26.46 ± 0.35 ^c,e^
ESP-MDGA1:2	2.96 ± 0.07 ^d^	10.65 ± 0.08 ^f^	23.23 ± 0.32 ^d^
ESP-MDGA1:4	3.32 ± 0.08 ^e^	11.52 ± 0.18 ^g^	28.91 ± 1.36 ^e^

* Values with the same lowercase letters (^a–g^) in the same columns did not show any significant difference at *p* < 0.05, according to Tukey’s HSD test.

**Table 4 foods-11-01922-t004:** Bioavailability, protein content, and solubility of powder samples.

Sample	Protein Content [%] *	Protein Solubility [%] *	Protein Digestibility [%] *
SP4	67.33 ± 0.59 ^f^	53.92 ± 0.65 ^g^	58.68 ± 0.31 ^a^
ESP4	64.71 ± 0.21 ^g^	63.51 ± 0.68 ^a^	60.47 ± 0.00 ^a^
SP-MD1:2	21.39 ± 0.23 ^a^	79.28 ± 0.39 ^b^	79.52 ± 1.79 ^b^
SP-MD1:4	12.57 ± 0.03 ^b,c^	79.06 ± 0.70 ^b^	79.72 ± 0.87 ^b^
SP-MDGA1:2	20.82 ± 0.24 ^a^	71.65 ± 0.00 ^c,d,e^	79.30 ± 1.69 ^b^
SP-MDGA1:4	12.46 ± 0.13 ^b,d^	65.02 ± 0.20 ^a^	82.73 ± 0.00 ^c^
ESP-MD1:2	18.19 ± 0.18 ^h^	73.35 ± 0.46 ^c,f^	90.12 ± 0.55 ^d^
ESP-MD1:4	11.65 ± 0.07 ^e^	82.88 ± 0.43 ^h^	73.62 ± 0.42 ^e^
ESP-MDGA1:2	18.92 ± 0.15 ^i^	72.04 ± 0.59 ^d,f^	84.23 ± 0.62 ^c^
ESP-MDGA1:4	12.02 ± 0.02 ^c,d,e^	69.82 ± 1.38 ^e^	77.09 ± 0.96 ^b^

* Values with the same lowercase letters (^a–i^) did not show any significant difference at *p* < 0.05 according to Tukey’s HSD test.

## Data Availability

Data are available on request from the author(s). The datasets generated during the experiment that support the findings of this study are available from the corresponding authors upon reasonable request.

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
