# Peer review of "Investigation of Bioavailability and Food-Processing Properties of Arthrospira platensis by Enzymatic Treatment and Micro-Encapsulation by Spray Drying"

_foods, 2022, doi:10.3390/foods11131922_

Round 1

Reviewer 1 Report

Abstract is OK.

Introduction

1- Line 64: Correct Spirulina to spirulina

2- Lines 86: Correct Spirulina to spirulina

Materials and Methods

1- Line 112: Why 4% (w/w)?

2- Line 113: Why 40 °C for 20 minutes?

3- Line 138: Why the inlet temperature was set to 120 °C?

4- Line 139: Why the resulting outlet temperature between 70 °C at a spray flow of 0.667 m3 /h?

Statistical Analysis

Line 244: Write full name before HSD

Results and Discussion

Line 278: Correct Spirulina to spirulina

Line 379: Write the year for Tan et al.

Line 396: Write the year and reference number [ ] for Tamm et al.

Figures are OK.

Tables

Tukey's test or Tukey's HSD test?

Conclusion is OK.

Author Response

Dear reviewer 1,

many thanks for the review of our manuscript. We have revised the manuscript taking your comments into account and prepared a cover letter for the point-by-point responses to your comments. Please, see the attachment.

Kind regards,

Patricia Maag

Reviewer 2 Report

The manuscript is interesting, but some correction is needed.

The sensory profile is not presented. We affirm in the aim that „... this study was to formulate a spirulina extract with reduced colour intensity, optimized thermal stability, increased solubility, moreover improved digestibility and sensory profile ...”

 Please verify the statistics. In Tables 1 and  4 some values are without lowercase letters (e.g., for some L*, a*, b* values).

Please use the term Equation instead of Formula in the entire manuscript.

L386: please use samples instead of references

Please explain under Table 2 the code for each sample.

Please revise the References section and write according to the

journal requirements.

Author Response

Dear reviewer 2,

many thanks for the review of our manuscript. We have revised the manuscript taking your comments into account and prepared a cover letter for the point-by-point responses to your comments. Please, see the attachment.

Kind regards,

Patricia Maag

Round 2

Reviewer 2 Report

The authors made the revisions and I consider that the article can be accept in the present form.